# An Assessment of Anthropogenic CO_2_ Emissions by Satellite-Based Observations in China

**DOI:** 10.3390/s19051118

**Published:** 2019-03-05

**Authors:** Shaoyuan Yang, Liping Lei, Zhaocheng Zeng, Zhonghua He, Hui Zhong

**Affiliations:** 1Key Laboratory of Digital Earth Science, Institute of Remote Sensing and Digital Earth, Chinese Academy of Sciences, Beijing 100094, China; yangsy@radi.ac.cn (S.Y.); hezhh@radi.ac.cn (Z.H.); zhonghui@radi.ac.cn (H.Z.); 2University of Chinese Academy of Sciences, Beijing 100049, China; 3Division of Geological and Planetary Sciences, California Institute of Technology, Pasadena, CA 91125, USA; zzhaoch@gmail.com

**Keywords:** anthropogenic CO_2_ emissions, GOSAT, atmospheric CO_2_ concentration

## Abstract

Carbon dioxide (CO_2_) is the most important anthropogenic greenhouse gas and its concentration in atmosphere has been increasing rapidly due to the increase of anthropogenic CO_2_ emissions. Quantifying anthropogenic CO_2_ emissions is essential to evaluate the measures for mitigating climate change. Satellite-based measurements of greenhouse gases greatly advance the way of monitoring atmospheric CO_2_ concentration. In this study, we propose an approach for estimating anthropogenic CO_2_ emissions by an artificial neural network using column-average dry air mole fraction of CO_2_ (XCO_2_) derived from observations of Greenhouse gases Observing SATellite (GOSAT) in China. First, we use annual XCO_2_ anomalies (dXCO_2_) derived from XCO_2_ and anthropogenic emission data during 2010–2014 as the training dataset to build a General Regression Neural Network (GRNN) model. Second, applying the built model to annual dXCO_2_ in 2015, we estimate the corresponding emission and verify them using ODIAC emission. As a results, the estimated emissions significantly demonstrate positive correlation with that of ODIAC CO_2_ emissions especially in the areas with high anthropogenic CO_2_ emissions. Our results indicate that XCO_2_ data from satellite observations can be applied in estimating anthropogenic CO_2_ emissions at regional scale by the machine learning. This developed method can estimate carbon emission inventory in a data-driven way. In particular, it is expected that the estimation accuracy can be further improved when combined with other data sources, related CO_2_ uptake and emissions, from satellite observations.

## 1. Introduction

Atmospheric carbon dioxide (CO_2_) is the most significant anthropogenic greenhouse gas (GHG) and its concentration in atmosphere has been increasing from 280 ppm since the preindustrial era to a level higher than 400 ppm at present at a global scale [1]. The enhancement of atmospheric CO_2_ has been known as one of the factors inducing global warming and playing an important role in climate change. Anthropogenic CO_2_ emissions, 70% of which come from fossil fuel combustion and industrial activities [2], are the main driver of the atmospheric CO_2_ concentration increase. If atmospheric CO_2_ concentration continues to increase at the current rate, 1.5 °C of global warming will be reached between 2030 and 2052, which will cause more climate extremes [3]. Atmospheric CO_2_ concentration, moreover, will be continually increasing as the rapid development of industrialization requires enormous energy around the world. In order to slow down the increase of atmospheric CO_2_ concentration, many countries are making efforts for CO_2_ emissions reduction. For that we need an efficient and reliable way to monitor CO_2_ emissions in order to evaluate the effectiveness of CO_2_ emissions reduction policy. 

Over the past 20 years, satellite-based measurements of greenhouse gases have been facilitating the way monitoring atmospheric constituents with the great advancement of satellite observing technology in the development of highly accurate sensors. It is also becoming the major data source to detect the change of atmospheric CO_2_ concentration at regional and global scales [4,5,6,7]. Compared with ground-based observation, CO_2_ concentration retrieved by satellite has global coverage and consistent observation characteristics, which can better reveal the spatio-temporal variation of atmospheric CO_2_ concentration. Currently, the GHG observing satellites in orbit include the Greenhouse gases Observing SATellite (GOSAT) from Japan, Orbiting Carbon Observatory 2 (OCO-2) from the USA and TanSat from China, which can provide us the column-averaged dry air mole fraction of CO_2_ (XCO_2_) dataset since 2009 [8,9,10].

Many previous studies indicated that XCO_2_ retrieved from satellite observations can detect changes of CO_2_ concentration induced by anthropogenic emissions [11,12,13]. The anthropogenic emission is expected to induce an increase of about 4 ppm of XCO_2_ around power plants [11]. With multi-year XCO_2_ dataset available from GOSAT and OCO-2, anthropogenic CO_2_ emissions have been quantified by excluding the background concentration. It was reported that megacities such as Los Angeles and Beijing, and high density urban regions such as eastern USA and the Beijing-Tianjin-Hebei area in northern China have about 2 ppm enhancements [14,15,16]. These studies mainly obtained regional CO_2_ enhancements in contrast to the background using empirical conversion factors. It has been shown that the XCO_2_ has a positive correlation with the anthropogenic CO_2_ emissions through correlating OCO-2 observations with emission inventories [17]. The correlation implies that satellite-based observations are capable to quantitatively assess the anthropogenic CO_2_ emissions through detection of XCO_2_ enhancements. Estimation of anthropogenic emissions from satellite-based observation can support the investigation of carbon emissions as a data-driven method, which is different to the conventional method in calculating emission inventory. Satellite observations can detect the CO_2_ changes in specific regions such as strong sources of anthropogenic emissions, e.g., megacities and high density urban area, so as to monitor CO_2_ emissions effectively. These studies, however, mostly focus on investigating enhancement of CO_2_ induced by anthropogenic emissions through regional contrast. It is still a challenge in using XCO_2_ data to quantitatively estimate the magnitude of anthropogenic CO_2_ emissions. This data-driven approach, as an additional way of quantifying anthropogenic CO_2_ emissions, can help policymakers to obtain more information for evaluating the effects for CO_2_ emissions reduction at both regional and global scales.

In this paper, we propose a method of using satellite-based observation to assess the anthropogenic CO_2_ emissions aiming to assist the national routine investigation of carbon emissions. We focus on mainland China as the studying area since it is a major national emitter of CO_2_ [18]. We extracted XCO_2_ anomalies (dXCO_2_) using XCO_2_ dataset obtained from GOSAT observations. The anomalies are found to be significantly correlated with anthropogenic CO_2_ emissions from the CO_2_ emitting sources such as power plant emission. We further introduce artificial neural network algorithm (ANN) to construct an estimation model for anthropogenic CO_2_ emissions based on the changes of atmospheric CO_2_ concentration derived from satellite observations.

## 2. Data and Methodology

### 2.1. XCO_2_ Retrievals and Mapping XCO_2_ Dataset

XCO_2_ retrieval data products from January 2010 to December 2015 are collected from GOSAT Atmospheric CO_2_ Observations from Space (ACOS v7.3). This dataset was produced by the Orbiting Carbon Observatory (OCO) team of the US National Aeronautics and Space Administration (NASA) using a full physics algorithm to retrieve XCO_2_ from GOSAT’s Spectrometer (TANSO-FTS) calibrated spectra measurements (Level 1B) [19]. In order to ensure high reliability of the data, only those data over land with high gain are used after screening and correction of systematic bias as described in the ACOS Level 2 Standard Product and Lite Data Product Data User’s Guide, v7.3 [20]. ACOS XCO_2_ retrievals have a standard deviation of error of 1.48 ppm when comparing with ground-based measures of Total Carbon Column Observing Network (TCCON) [21]. Figure 1 shows the total counts of XCO_2_ data points during 6 years from January 2010 to December 2015, and their temporal variation in the study area. An annual increase of XCO_2_ and the seasonal variation can be clearly seen. 

However, XCO_2_ data are irregularly distributed and have many gaps in space and time as shown in Figure 1a because of the limitation of GOSAT observation mode, cloudy and data screening. To investigate the space-time changes of XCO_2_, we generate a mapping XCO_2_ dataset in which those gaps are filled using the kriging interpolation method based on the spatio-temporal geo-statistics model [22,23,24]. The mapping XCO_2_ dataset is generated mainly in Chinese mainland area from 18° N to 57° N and from 65° E to 148° E with 0.5° × 0.5° grid cells and 3-day interval in time from 1 January 2010 to 31 December 2015. In order to match with collected ODIAC emission dataset in 1° × 1°, we resampled the spatial resolution of data we used in this paper to 1° × 1°. This mapping dataset is hereafter referred to as Mapping-XCO_2_.

### 2.2. Anthropogenic Emission Data

We collected two datasets of the bottom-up anthropogenic CO_2_ emissions. One is the Open-source Data Inventory for Anthropogenic Carbon dioxide (ODIAC) for same years as the used XCO_2_ dataset in this study. The other is the CARbon Monitoring for Action (CARMA) power plant database in 2009. The specifications of these data are described in Table 1.

The ODIAC emissions data product is a global 1° × 1° gridded monthly fossil fuel CO_2_ emission inventory, developed based on country level fossil fuel CO_2_ emission estimates, fuel consumption statistics, satellite-observed nightlight data, and point source information (geographical locations and emission intensities) from the CARbon Monitoring for Action (CARMA) power plant database (ODIAC2015a, available at http://db.cger.nies.go.jp/dataset/ODIAC/). The global nightlight data were used as a geo-referenced, spatial proxy to determine the spatial extent of anthropogenic emissions from line and diffused (area) sources (e.g., road traffic, residential or commercial fuel consumption) [25]. The ODIAC gridded emissions fields defined on a global rectangular (latitude ×longitude) coordinate are remapped to meet the grids resolutions for each simulation domain.

Additionally, the CO2 emissions from power plant, which is one of the dominant CO_2_ emitting sources, are collected in the study area from the database of Carbon Monitoring for Action (CARMA, available at http://carma.org/plant). At the same time, we unify the units of the two sets of emission data to ton, and take the logarithm of two emission data base on 10 (refer to as lgE) to facilitate the calculation.

### 2.3. Methodology

The method for estimating anthropogenic CO_2_ emission include three major steps as shown in Figure 2. 

Firstly, we enhance the signals of CO_2_ from anthropogenic emission in XCO_2_ which is described in Section 2.3.1. Secondly, we apply the training datasets of XCO_2_ and ODIAC in 2010–2014 to GRNN to get the estimating model of anthropogenic emission which is described in Section 2.3.2 in detail. Thirdly, anthropogenic emissions are estimated by GRNN model using XCO_2_ in 2015, and validated by comparing with ODIAC data in 2015.

#### 2.3.1. Variable of XCO_2_ Used for Estimation of Anthropogenic Emission

The magnitude of XCO_2_ include CO_2_ emitted by anthropogenic activities, the fluxes of terrestrial biosphere, fluxes transported by atmospheric wind fields [27,28] and CO_2_ of regional background. We introduce therefrom an interannual variability by removing the regional background signal and calculating their annual mean to enhance the signals of CO_2_ from anthropogenic emission as following equation proposed by Hakkarainen et al. [17]:
(1)$$dXCO_{2}\left(grid,t\right)=XCO_{2}\left(grid,t\right)-MXCO_{2}\left(t\right)$$where dXCO_2_ (grid,t) indicates the deviation from regional background for each grid at a specific time unit t where t is the 3-day unit of used mapping-XCO_2_ data; XCO_2_ (grid,t) is XCO_2_ for each grid at time t from mapping-XCO_2_ data; MXCO_2_(t) is median of XCO_2_ for all girds in the study region at time t calculated from mapping-XCO_2_ data with 0.5° × 0.5° grid cell. Lastly we apply the annual mean of dXCO_2_ (grid,t) for the year from 2010 to 2015 in the estimation of anthropogenic emission. This annual mean of dXCO_2_ (grid,t) could detrend the seasonal variation at locale and simultaneously reduces the effect of the atmospheric transport [17].

We computed the monthly averaged dXCO_2_ and annual averaged dXCO_2_ for each grid to generate monthly averaged dXCO_2_ dataset and annual averaged dXCO_2_ dataset from the year 2010 to 2015 with 1° × 1° grids using the mapping XCO_2_ dataset from 2010 to 2015. The annual dXCO_2_ dataset and ODIAC data will be used in the following analysis.

#### 2.3.2. Estimation of Anthropogenic CO_2_ Emission by Neural Network Development

Because XCO_2_ variations are forced by anthropogenic emissions, exchange between the atmosphere and the ocean and the terrestrial biosphere [27,28], there are both non-linear and linear mapping between XCO_2_ and emissions. Here we adopt a General Regression Neural Network (GRNN) algorithm [29] to represent non-linear mapping between the independent variables (dXCO_2_ in this study) and dependent variable (CO_2_ emissions in this study). GRNN directly draws the function estimate approximating any arbitrary function between the input and output vectors of variables. The GRNN converges to the optimal regression result when the training samples increases in number, meanwhile, the error of estimation is closed to 0. There are four layers in the GRNN model we used, an input layer, a hidden layer, a summation layer, and a decision layer (Figure 3; [30,31]). In the input layer, each neuron corresponds to an independent variable which is defined as a mathematical function, the independent variable values will be standardized. Then the standardized independent variable values were transferred to the neurons in the hidden layer. In this layer, each neurons stores the values of the independent variables and the dependent variable, and a scalar function will be calculated. There are two neurons in the summation layer, the denominator summation unit sums the weight values coming from the hidden neurons, and the numerator summation unit sums the weight values multiplied by the actual target dependent variable value for each hidden neuron. At last, dividing the value accumulated in the numerator summation unit by the value in the denominator summation unit in the decision layer, we uses the division result as the predicted target dependent variable value [32].

According to the calculation steps of developing a neural network, we need to standardize all the independent and dependent training variables, so that in the input layer all training data will have the same order of magnitudes.
(2)$$d\left(x_{0}-x_{i}\right)=\overset{p}{\underset{j=1}{\sum }}{\left[\frac{x_{0j}-x_{ij}}{\sigma }\right]}^{2}$$where p denotes the dimension of variable vector $x_{i}$, σ is the spread parameter, whose optimal value is determined by minimizing the root mean square error (RMSE) between the training data and the predicted values of the dependent variable. 

The weight of the denominator neuron is set to 1.0. The GRNN training algorithm uses only one adjustable parameter σ for a given training set. Here we use “the holdout method” [29] to optimize the σ value, and detailed introduction can refer to the article [29]. The predicted target dependent variable, the ODIAC CO_2_ emissions, is defined by the following Equation (3):
(3)$$\hat{y}\left(x_{0}\right)=\frac{\sum_{i=1}^{n}y_{i}e^{-d\left(x_{0},x_{i}\right)}}{\sum_{i=1}^{n}e^{-d\left(x_{0},x_{i}\right)}}$$where the values calculated with the scalar function in a hidden neuron i are weighted with the corresponding values of the training samples $y_{i}$, and then passed to the numerator neuron. n is the number of training samples.

## 3. Results and Discussion

### 3.1. Estimated Anthropogenic Emissions by GRNN

We use the annual dXCO_2_ dataset and ODIAC CO_2_ emissions data from the year 2010 to 2014 as the training dataset, which have the total of 5415 samples available, to build a GRNN model for estimating anthropogenic emission. By applying “the holdout method” described in Section 2.3, we obtain the optimized spread parameter σ as 0.1. Then we apply GRNN model to the annual dXCO_2_ data in 2015 to predict target dependent variable, anthropogenic emission with the same unit as the ODIAC CO_2_ emissions.

The CO_2_ emissions estimated using the annual dXCO_2_ and the actual ODIAC CO_2_ emission in 2015 are shown in Figure 4. Comparing Figure 4a with Figure 4b, we can see that the spatially changing pattern of estimated emission by satellite-based observation is exactly similar as that of the actual magnitude of ODIAC. Moreover, the estimated emission presents a more smoothing spatial details than the actual emission, which is mainly because the Kriging procedure smooths the CO_2_ signals from point sources of strong anthropogenic emission, and 10 km spatial resolution of each GOSAT footprint observations also smooths the signals. The magnitude of estimated emission is generally less than that of ODIAC. Figure 5 presents the differences between them and the corresponding histogram.

It can be seen from Figure 5a that the difference between the estimated CO_2_ emission and ODIAC emission mainly change from −5 Mt to 5 Mt, which accounts for 91% of the total grids. The magnitude of difference from −1 Mt to 1 Mt accounts for 71% of the total grids. The low magnitude of ODIAC emissions in the range of 1–10^4^ t/year shown in Figure 4b are generally underestimated by satellite-based observations (shown in yellow in Figure 5a). These are mostly located in semi-arid grasslands, forests in the northern areas as shown in land use map of Figure 5c. 

This underestimation implies that the emission estimated by dXCO_2_ has high uncertainty in the areas of low anthropogenic emission that is likely due to the CO_2_ uptake of biosphere which is still remaining in dXCO_2_. The estimated emission, moreover, is much lower than ODIAC emission over the areas around big cities, such as Beijing, Shanghai, Guangzhou. This underestimation indicates that the smoothing effects of the estimated emission, which is likely because the spatial resolution of GOSAT observations (10 km) is not sufficient to detect the emission of point sources. On the other hand, the estimated emissions are generally larger than ODIAC emission in the south-eastern region of China where there are many anthropogenic emitting sources which can be seen in Figure 8. The general overestimation in this region is likely because the large emitting sources around raise the concentration of CO_2_ over those non-emitting areas nearby them through the atmospheric transport. 

Lastly, comparing the satellite-based estimation of CO_2_ emissions with ODIAC emission for all grids as shown in Figure 6, we find they show a significant correlation (R^2^) of 0.65 with p value less than 0.01.

### 3.2. Discussion of Correlation between Retrieved X_CO2_ and Anthropogenic Emissons

It has been indicated that the cluster of XCO_2_ changes derived from GOSAT observations shows a correlating coefficient of 0.5 with anthropogenic emission. This correlation is more significant than a single grid of XCO_2_ as the atmospheric CO_2_ measurement is an instantaneous snapshot of the realistic atmosphere [33]. Its clustering analysis is derived from original XCO_2_ data. 

We segment the ODIAC emissions which are binned according to every 0.3 t/yr of lgE (Figure 7a) using mean emission calculated from annual emission during 2010–2015, and then make correlation analysis between the mean of emission and mean of dXCO_2_ within binned regions. It is found that the segmental mean of dXCO_2_ demonstrate a significant and positive correlation with ODIAC emissions in which the determined coefficient (R^2^) for all data is up to 0.82 (Figure 7b) and the dXCO_2_ demonstrate strong positive linear correlation with emission starting from 10^4^ t/yr where R^2^ is up to 0.95 (red line in Figure 7b). The dXCO_2_ is almost unchanged in the region with emission lower than 10^4^ t/yr. These results imply that satellite observations of atmospheric CO_2_ could be used to estimate regional anthropogenic emissions for those regions with larger magnitude of anthropogenic CO_2_ emissions. Additionally, we overlay the CARMA power plants dataset on the mean dXCO_2_ from the annual dXCO_2_ during 2010 to 2015 (Figure 8a). It can be seen that the high dXCO_2_ are corresponding to high-density power plants, especially in northeast China. We accumulate the magnitude of emissions of power plants within one grid of mapping XCO_2_ dataset, then we segment emissions of power plants which are binned according to every 0.3 t of lgE, and take correlation analysis between the mean of power plants emission and the mean of dXCO_2_ within binned regions (Figure 8a). dXCO_2_ demonstrate strong positive linear correlation with power plants emission starting from 10^6^ t (blue dots). The grids they represent are distributed consistently with high dXCO_2_ area. The result demonstrates a R^2^ of 0.59 which is less than regional statistics. Power plants emission lower than 10^5.5^ t demonstrate weak linear correlation with dXCO_2_ because the influence of CO_2_ uptake of biosphere.

From Figure 8a, it can be found that the dXCO_2_ in western area, the desert area of Xinjiang, shows high values even if there are much less anthropogenic emission over this area as shown in Figure 4b. This is likely resulted in the uncertainty of ACOS XCO_2_ retrievals in desert which has been indicated by Bie et al. [34].

## 4. Conclusions

In this paper, to support the verification of bottom-up inventory of anthropogenic emission, an anthropogenic CO_2_ emission estimation method using a machine learning technique is applied to the gap-filled ACOS XCO_2_ dataset over the mainland of China derived from GOSAT observations. The annual emission signatures, indicated by dXCO_2_, is enhanced by removing the background XCO_2_ from the 2010 to 2015 XCO_2_ data. We then apply the annual averaged dXCO_2_ from 2010 to 2014 to build an estimating model of anthropogenic emission using an artificial network approach. The model is verified by estimating results in 2015 and comparing with the ODIAC emissions. Lastly, we quantify the correlation between the annual dXCO_2_ and the magnitude of anthropogenic emission. Our result indicate that the anthropogenic emission can be estimated at regional scale by the changing magnitude of XCO_2_ especially for those regions with larger emissions. However, it has relatively higher uncertainty to grasp the CO_2_ signals of the low or without anthropogenic emission areas and point emitting sources. The CO_2_ uptake of biosphere and fluxes of wind field affect the estimation when using the annual dXCO_2_. The observation mode of GOSAT satellite in space and time and fast mixing of atmospheric CO_2_ also affect the detection of point emitting sources.

Our study demonstrates that the XCO_2_ derived from satellite observation can effectively provide a way to reveal the spatial patterns of underlying anthropogenic emissions. It is expected that the estimation of anthropogenic emission could be greatly improved by using more and more XCO_2_ data from multi-satellite such as OCO-2, OCO-3, GOSAT-2, and TanSat in future. Moreover, we can combine the ancillary data related with CO_2_ uptake and emission which can be obtained by satellite remote sensing observations at the same time, such as gross primary production (GPP), industrial heat source from VIIRS (Visible infrared Imaging Radiometer) Night fire product for point sources, Night light from Defense Meteorological Satellite Program/Operational Linescan System (DMSP/OLS), to constrain the estimating model developed in this study. This data-driven approach based on satellite-based observations can offer the possibility of rapid updates for anthropogenic CO_2_ emissions, and provide a new way of investigating anthropogenic emissions to support the implement of regional reduction of carbon emissions.

## Figures and Tables

**Figure 1 sensors-19-01118-f001:**
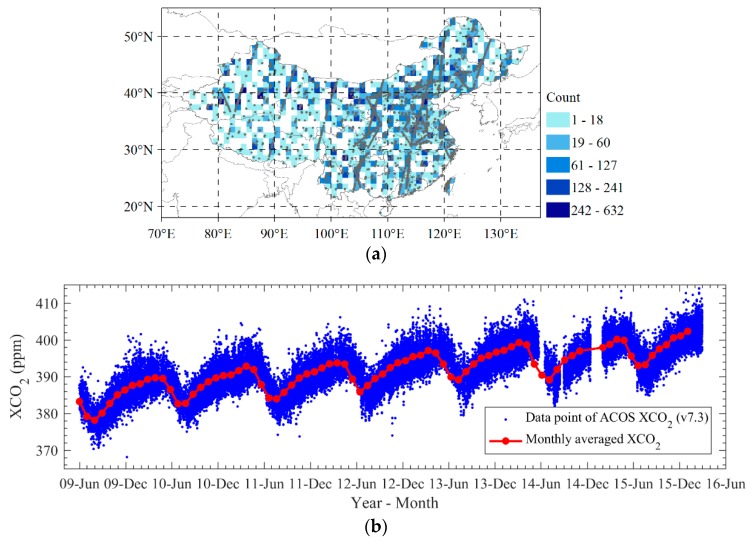
Number of available retrievals and temporal changes of collected XCO_2_ data: (**a**) the number of XCO_2_ data points within 1° × 1° grid for 6 years from 2010 to 2015; (**b**) the temporal variation of XCO_2_ and the monthly averages.

**Figure 2 sensors-19-01118-f002:**
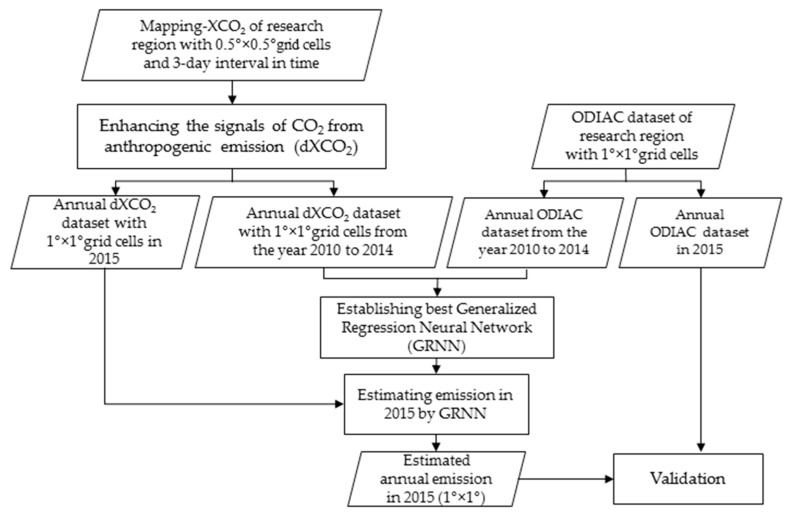
Flowchart of estimating anthropogenic emission using XCO_2_ data obtained by GOSAT observations. It consists of three major steps, firstly enhancing the signals of CO_2_ from anthropogenic emission in XCO_2_; secondly establishing GRNN model using the training datasets; the last estimating the anthropogenic emissions and validating the result.

**Figure 3 sensors-19-01118-f003:**
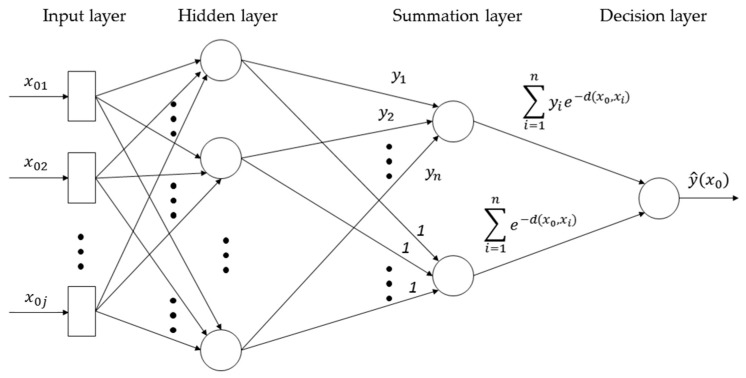
Schematic diagram of the generalized regression neural network architecture based on Cigizolu and Alp [30].

**Figure 4 sensors-19-01118-f004:**
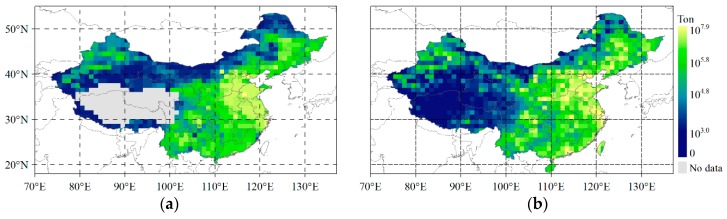
The anthropogenic CO_2_ emissions in 2015 in China: (**a**) CO_2_ emission estimated using GRNN based on the annual dXCO_2_ in 2015 from GOSAT observations. The Tibet area shown as blank is filtered due to their high uncertainty XCO_2_ retrievals; (**b**) the CO2 emission from ODIAC in 2015.

**Figure 5 sensors-19-01118-f005:**
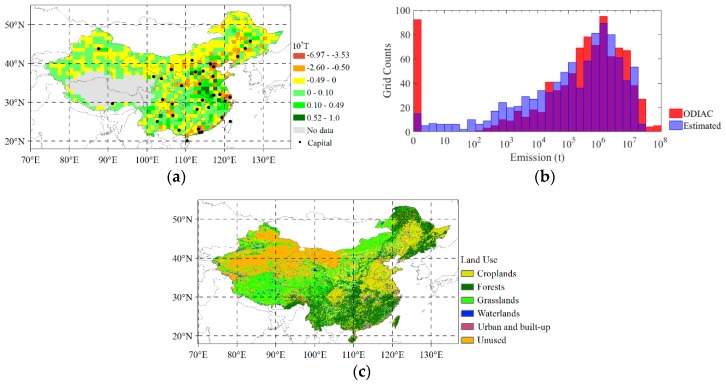
(**a**) The difference between the estimated CO_2_ emission using mapping-XCO_2_ in 2015 and ODIAC emission in 2015; (**b**) Histogram comparison of the estimated and the ODIAC CO_2_ emission (in unit of Ton/year) in 2015; (**c**) Land use of China in 2010.

**Figure 6 sensors-19-01118-f006:**
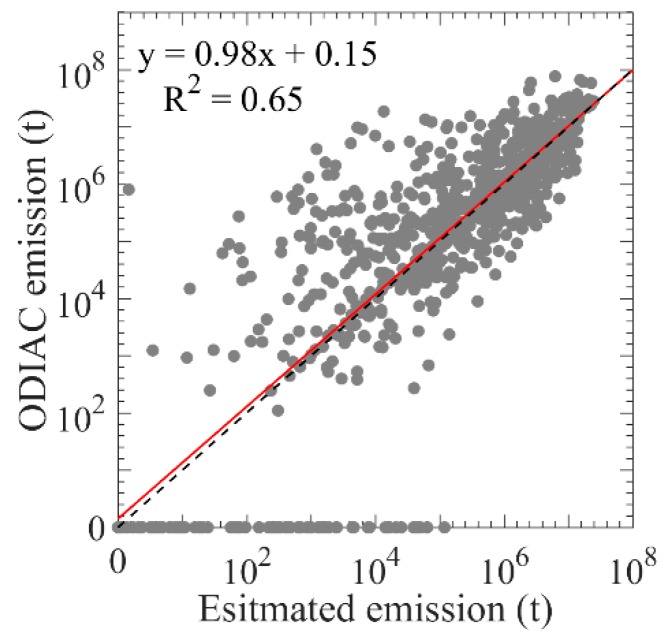
Scatterplot between estimated CO_2_ emission using mapped-XCO_2_ data and the actual ODIAC emission in 2015, in which red line is the regression line between estimated emission and ODIAC emission, and black dotted line is the 1:1 line.

**Figure 7 sensors-19-01118-f007:**
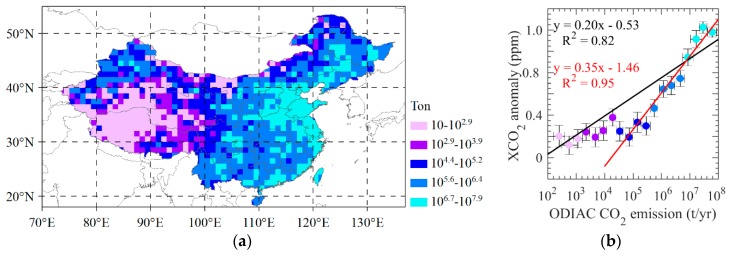
(**a**) Segment of ODIAC emissions, where the data are binned by every 0.3 t/yr of lgE using mean emission calculated from annual emission during 2010–2015; (**b**) correlation between mean ODIAC CO_2_ emissions and mean dXCO_2_ calculated from annual dXCO_2_ during 2010–2015 for each segment, where red line is the regression line between dXCO_2_ and ODIAC CO_2_ emissions with emission lager than 10^4^ t/yr.

**Figure 8 sensors-19-01118-f008:**
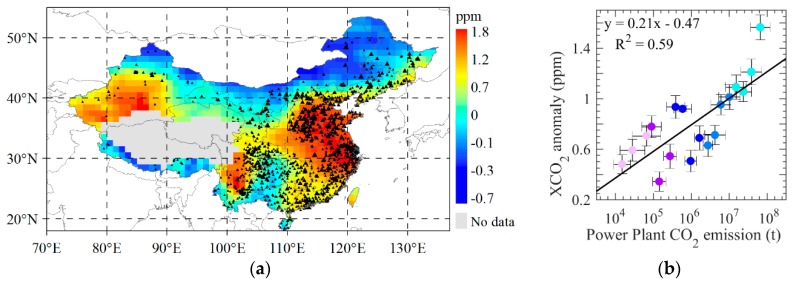
(**a**) the mean of dXCO_2_ from the annual dXCO_2_ during 2010 to 2015 overlaid with CARMA power plants locations; (**b**) the correlation between emission of CARMA power plants within 1° × 1°grid which are binned by every 0.3 t/yr of lgE and the corresponding mean dXCO_2_, different color dots represent different segment of CARMA power plants emissions.

**Table 1 sensors-19-01118-t001:** Basic specifications of ODIAC and CARMA bottom-up CO_2_ emission datasets.

	ODIAC	CARMA
**Grid, timely unit/period**	1° × 1°, Month/2010–2015	Points/2009
**Unit**	Ton	Ton
**Statistical sectors**	Point sources non-point sources	-
	Cement production	
	Gas flaring	
	International aviation and marine bunker	
**Used data sources**	Fuel statistic data published as united nation energy statistics database BP statistical review of world energy 2017	The environmental protection agency and department of energy International atomic energy agency
**Producer**	Center for global environment research, national institute for environment studies	Center for Global Development
	Oda, T et al. [25]	Wheeler, D et al. [26]

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
