# Peer review of "An Assessment of Anthropogenic CO2 Emissions by Satellite-Based Observations in China"

_sensors, 2019, doi:10.3390/s19051118_

Round 1
Reviewer 1 Report
Please refer to my comments described in a separated file.

Author Response
Dear Editor and Reviewers,
We are truly grateful to your help and reviewers for giving critical comments and thoughtful suggestions. The valuable comments not only helped us with the improvement of our manuscript, but also suggested some ideas for future studies.
We have made careful modifications on the original manuscript according to all the comments and suggestions from the reviewers, and given item-to-item responses to each reviewer. The main changes include:
(1) We added flowchart (Figure 2) and Section 2.3 to briefly explain the principle of the method, and change the original Section 2.3 and Section 2.4 to Section 2.3.1 and Section 2.3.2.
(2) We revised several vague descriptions to make sure the revised manuscript is easier to understand.
(3) We corrected some mistakes in figure or section, and adjusted the serial number of the reference.
(4) We changed the order of project funds, and put project “Key Research Program of the Chinese Academy of Sciences” (Grant No. ZDRW-ZS-2019-1) at the first place. We changed and confirmed the English name and Grant number of the first project this time.
(5) We corrected the English phrasing that are lack of clarity pointed or un-pointed out by reviewers.
The revised manuscript and responses to the reviewers have been uploaded to the website.
All the changes made to the manuscript were highlighted in yellow.
Sincerely,
The Authors

Reviewer 2 Report
The overall premise of this paper deals with estimating changes in anthropogenic CO2 emissions using a combination of satellite observations, neural networks and machine learning methods. Two datasets of the bottom-up anthropogenic CO2 emissions were collected. One is the Open-108 source Data Inventory for Anthropogenic Carbon dioxide (ODIAC) for same years as the used XCO2 109 dataset. The other is the CARbon Monitoring for Action (CARMA) power plant database in 2009.
The paper and approach adopted are original and I suspect this is a publishable paper. I have a few comments that should be addressed before the paper is accepted.
1) I have spotted a few typos and mistakes in the various sections of the paper and I suggest that the paper is given to someone with English as their mother tongue in order to ensure that these are addressed
2) I recommend that you mention the IPCC's latest report on 1.5 C in the introduction as this is currently high on the agenda
Author Response

(The authors gave the same response as above.)
